# Recognition of Coat Pattern Variation and Broken Tail Phenomenon in the Asiatic Golden Cat (*Catopuma temminckii*)

**DOI:** 10.3390/ani12111420

**Published:** 2022-05-31

**Authors:** Yuan Wang, Dajiang Li, Pubu Dunzhu, Wulin Liu, Limin Feng, Kun Jin

**Affiliations:** 1Ecology and Nature Conservation Institute, Chinese Academy of Forestry, Beijing 100091, China; wangyuan@caf.ac.cn; 2Research Institute of Natural Protected Area, Chinese Academy of Forestry, Beijing 100091, China; 3Key Laboratory of Biodiversity Conservation of National Forestry and Grassland Administration, Beijing 100091, China; 4Forestry Inventory and Planning Institute of Tibet Autonomous Region, Lhasa 850000, China; ldjxzlgy@163.com (D.L.); pubudunzhu0430@163.com (P.D.); wulinliu01@163.com (W.L.); 5Institute of Ecology, Beijing Normal University, Beijing 100875, China; fenglimin@bnu.edu.cn

**Keywords:** Asiatic golden cat (*Catopuma temminckii*), coat morph, recessive gene, camera trap, Yarlung Zangbo Grand Canyon National Nature Reserve, broken tail phenomenon

## Abstract

**Simple Summary:**

A variety of new survey technologies are continuously being developed and used in wildlife monitoring. Rapidly advancing and widely used camera trap survey technology has helped to capture data and gain insights into many species. The Eastern Himalayas is a global biodiversity hotspot with exceptionally high species diversity. The Asian golden cat is widely distributed in the Yarlung Zangbo Grand Canyon National Nature Reserve. It inhabits seasonal rain forests from 100 m above sea level to the Rhododendron forest up to 3500 m above sea level. Coat pattern variation in the Asian golden cat is particularly prominent in this region. The common color type is the most widely distributed, followed by nine other types. We found 10 coat pattern variations and two coat patterns with a broken tail made up 0.32% of independent photos taken during a long-term nine-year monitoring program. The variation in coat patterns is indicative of the geography of the region. Environmental conditions regulate and activate the genetic diversity of Asian golden cat phenotypes. This study further strengthened the understanding of the basic knowledge of golden cat color types and lays the foundation for exploring the diversity of golden cat color types at the molecular level.

**Abstract:**

The Asian golden cat (*Catopuma temminckii*) is the most varied wild cat species in terms of coat color. Understanding coat pattern variation will help to elucidate the mechanisms behind it as well as its relationship with the environment. We conducted long-term (2013–2021) monitoring of Asian golden cats in the Yarlung Zangbo Grand Canyon National Nature Reserve, Tibet, using camera traps at 283 points over 89,991 camera days. A total of 620 cat photos were recorded, including 344 (55.48%) with recognizable color patterns. Vector graphics of the coat patterns were extracted from the field image data, which revealed 10 color types in the ratio common: cinnamon: reddish-brown long hair: ocelot: blackening: melanistic: gray: brown: brown short hair: pure black = 123:76:57:35:22:8:7:7:5:4. The genes for coat pattern variation are widespread in the Asian golden cat population and are relatively stable. The increase in population size intraspecific competition has led to the tail break phenotype in individual cats. The gene encoding for tail breakage in Asian golden cats remains unknown. This study provides basic information for understanding faunal diversity in the Eastern Himalayan biodiversity hotspot and serves as a reference for studies on the formation mechanisms for feline color pattern diversity.

## 1. Introduction

Interpretation of the diversity of fur color patterns requires an understanding of the mechanisms of color development and their ecological adaptive value [1]. The feline coat pattern and the complexity and irregularity of the pattern properties may be related to the habits, physiological regulations, and information exchange of the feline [1,2,3]. Variations in coat coloration are found in many mammalian groups, and spots, stripes, and other markings play an important role in camouflage, predator avoidance, and group social interactions [4,5,6].

Gene control of the transition between different pigmentations may be due to changes in the activity of different enzymes between multiple molecular levels and shifts between multiple stable editing patterns [3,4]. This implies that small changes in the level of external regulators may lead to substantial changes in the proportion of pigment colors produced and could explain the clearly delineated black or red coloration of mammalian coat patterns [7]. Theoretical studies have proposed mathematical and physical models of hair coat texture, which can provide a basis for revealing the mechanisms of hair coat texture formation in mammals [7,8,9,10]. One of the models gaining approval has been developed using the reaction-diffusion theory, which simulates the fur patterns of animals. This model explains the connection between feline fur patterns and ecology in terms of the endogenous aspects of cause and development, and the exogenous aspects of adaptation and evolution [1,9].

Wild cats (*Felis catus*) were domesticated in the Middle East approximately 9500 years ago and worldwide soon after. The coat of domestic cats exhibits a wide range of colors (single or a mixture of colors) and pattern variation (spots and stripes) [11,12,13,14,15]. This wide variety of phenotypes results from multi-locus gene interactions, such as those encoding black [16], brown and cinnamon [17], pale [18], albino [19,20], long hair [17,19,20,21], orange [3,5], and silver [22]. Variable coat coloration is a common polymorphism in felines, reaching high population frequencies in some cases, but is never completely fixed [5]. The process of fur diversification can be observed in most mammals undergoing domestication. This is often because populations of domesticated animals are no longer subjected to natural selection but are artificially selected by humans for preferred colors and patterns [3,23]. The chromotypic polymorphisms on the body of domestic cats, including spots, patches, stripes, and color variations, are similar to those of the wild Asian golden cats (*Catopuma temminckii*) and are effective for use in cross-referencing studies and as supporting evidence.

The Asian golden cat is a medium-sized cat with a body length of 75–100 cm, a tail length of 35–60 cm, and a weight of 8–16 kg. It is one of the representative species of small and medium-sized cats in China [24,25,26,27]. The origin of the specimen type of the golden cat is Sumatra. Its distribution area extends westward from South China and Southwest China to the southern foothills of the Himalayas and southward to Southeast Asia, including China, Vietnam, Laos, Cambodia, Thailand, Malaysia, Indonesia, Myanmar, India, Menggala, Bhutan, and Nepal [26]. In China, the golden cat has been historically distributed widely in east, central, south, and southwest China, but its distribution scope has shrunk sharply during the last half-century. The existing range may have been highly fragmented and island-like [27]. In recent years, confirmed records of domestic golden cats show that they are only found in northern and western Sichuan, southern Shaanxi, western and southern Yunnan, as well as some areas of southeastern Tibet [27].

Different color patterns of the Asian golden cat, its spatial distribution, and social interactions have been studied [11,12,13]. However, sampling biological material from the Asian golden cat has been challenging. Studies on the formation mechanisms and distribution of color patterns in this species are limited [11], and molecular studies on the fur color pattern are lacking. Long-term monitoring of the Asian golden cat revealed a relatively large variation in the color pattern that is very similar to that of domestic cats. The contributing factors to such a large variation in the Asian golden cat and domestic cat compared to nearly 40 feline species worldwide remains unknown [14]. Domestic cats are relatively easy to sample, and the genetic control of the coat color in domestic cats is well understood. Therefore, we aimed to explain the color change in golden cats through the mechanism of color change in domestic cats. Our study is a mapping hypothesis of similar phenomena between different species of the same genus in the Felidae, which is due to the difficulty in studying the molecular mechanisms of different color types of golden cats. At present, there is considerable doubt around the understanding of color variations in golden cats.

In this study, the Asian golden cat was surveyed in the Yarlung Zangbo Grand Canyon National Nature Reserve in Tibet. Nine years of camera-trap monitoring of the species revealed ten fur color types. We aimed to address the following questions: (1) How many different color types exist in the wild Asian golden cat population in the region? (2) Can existing monitoring data from this species explain the numerous color type variations? (3) Can the mechanisms of color pattern variation and the biological significance of the variation in domestic and Asian golden cats be explained? The results of this study will provide a baseline reference and basic data for research and conservation management of this species in protected areas.

## 2. Materials and Methods

### 2.1. Study Area

The Yarlung Zangbo Grand Canyon National Nature Reserve (hereinafter referred to as the Grand Canyon Reserve; 29°5′48″–30°20′7″ N, 94°45′28″–96°5′39″ E) is in Linzhi City in southeast Tibet. The administrative reserve area covers an area of 9168 km^2^ that encompasses 14 townships in the Bayi District, Milin County, Bomi County, and Medog County (Figure 1).

At the mouth of the Yarlung Zangbo River, at the southern end of the reserve, there is an altitude difference of more than 7500 m from the Namcha Barwa peak in Pasighat, with altitudes of 154–7782 m, which forms a large gap within a horizontal distance of 40 km. This difference in altitude results in an extreme vertical distribution of climate. The area is specifically divided into seven climatic zones, viz., a humid climate zone (500–1000 m) in the northern margin of low mountain tropics, a mountainous subtropical semi-humid climate zone (1000–2400 m), a sub-alpine semi-humid climate zone (2400–3200 m), a sub-alpine cold and warm climate zone (3200–4000 m), an alpine cold climate zone (4000–4300 m), an alpine freezing weathered zone (4300–4800 m), and an alpine freezing climate zone (4800–7782 m) [11].

Due to its size and unique topography, climate, and vegetation, the Grand Canyon Reserve features rich biodiversity. The survey data identified 13 species of national I key protected wild animals and 36 species of national II key protected wild animals in the reserve, including nine species of wild cats, namely the Bengal tiger (*Panthera tigris tigris*), snow leopard (*P. uncia*), leopard (*P. pardus*), clouded leopard (*Neofelis nebulosa*), lynx (*Lynx lynx*), Asian golden cat, cloud cat (*Pardofelis marmorata*), leopard cat (*Prionailurus bengalensis*), and jungle cat (*Felis chaus*) [11].

### 2.2. Camera-Trap Survey Method

Fourteen representative areas of different elevation bands and vegetation types were selected in the reserve as survey areas. Camera traps were used to detect and record the wild Asian golden cats in each area. The survey was conducted over six time periods: January 2013–July 2013, October 2013–May 2014, October 2016–April 2017, October 2017–July 2018, August 2018–November 2019, and July 2020–December 2021. A total of 283 camera-traps survey sites were deployed, covering an elevation span of 582–3479 m, with a cumulative survey workload of 89,991 camera days (Table 1).

Infrared cameras (Acorn Ltl 6210; Ltl Acorn, Desa Moineis, IA, USA) were set to continuously shoot three photographs and one video of 15 s after each trigger. The sensitivity was set to medium, and the photo imprint was turned on. No physical concealment was conducted during camera installation, and no food bait or attractant was placed nearby. For the same camera position and similar time or similar camera position and similar time, consecutive photographs of Asian golden cats were taken to compare body size, side, spotting, coat color, and other physical characteristics as well as animal behavior to determine whether they belong to the same individual. Photographs taken at different times were defined as independent photographs (IP). (1) When an Asian golden cat was photographed by an infrared camera at a single site, it was recorded as an effective detection of the cat (where an “effective detection of the cat” is defined as one independent effective photo). (2) Within 30 min of obtaining the first photo, any further photos of the same species (whether it was the same individual or not) continuously taken at the same site were counted under the same detection. The photos taken in this time interval are referred to as the number of independent effective photos. (3) The number of independent effective photos is not related to the number of individuals of the same kind of animal taken in a single photo or during a single detection [11].

### 2.3. Camera-Trap Data Processing

#### 2.3.1. Camera-Trap Data Identification

Based on previous experience with long-term field surveys and deployment of the camera traps, only photos from the same camera with good daytime imaging and visualization were analyzed. Photos or videos with poor nighttime imaging were disregarded to reduce errors caused by misidentification. Pictures taken using different cameras were used to verify the body color of the animal.

#### 2.3.2. Determination of Coat Pattern Variation among Asian Golden Cats

Body shape, overall fur color, behavior, limb color, tail color, and the facial pattern of the cats were observed and compared. The color, spots, and patterns on the head, ears, back, limbs, tail, and other body parts of different individual golden cats were recorded and compared. We classified the body color of Asian golden cats according to the different color systems controlled by eumelanin and pheomelanin, that is, eumelanin controls the black pigmentation, and pheomelanin controls the red pigmentation [7].

Artrage 6.0 painting software, Ambient Design, New Zealand. (accessed on 9 December 2021, https://www.artrage.com/), GIMP, University of California Berkeley, Berkeley. (GNU Image Manipulation Program) version 2.10 (accessed on 9 December 2021, http://gimp.baisheng999.com/), and Photoshop 2021, Adobe, America. (accessed on 27 April 2022, https://www.adobe.com/) were used to process pre-selected photo images, adjust the parameters that affect the color type evaluation, such as color white balance and exposure, to the same standards, and to pass the color guide. The color type was determined for each Asian golden cat photographed, and vector images of the different color types were created at a 1:10 scale.

## 3. Results

### 3.1. Camera Trap Results

The Asian golden cat was recorded in 12 survey areas and no individuals were recorded on the south bank of the Yajiang River and Uma Mountain. The lowest altitude of the recorded sites was 815 m, and the highest altitude was 3479 m. A total of 620 independent photos of Asian golden cats were obtained (Table 1). In the surveyed altitude range (582–3479 m), other identifiable cats in the ‘bycatch’ sympatric distribution included the leopard cat, cloud cat, and clouded leopard.

### 3.2. Color Type Categories and Proportions

Among the 620 independent photos of Asian golden cats obtained in this survey, the color type could be identified in 344 photos, accounting for 55.48% of the total number of photos of Asian golden cats. These photographs were used for the identification of the color types. Ten different color patterns were identified: cinnamon, brown, common, ocelot, reddish-brown long hair, short brown hair, gray, blackening, melanistic, and pure black (Figure 2) [11,13].

The external morphological features shared by individuals of all color types were: facial markings relatively consistent, with a wide white stripe at the inner corner of each eye, followed by a brown stripe that extends across the forehead to the back of the head; brown stripes accompanied by fine black stripes on both sides; eyebrow lines white, varying in width; a white line on each side of the cheek extending diagonally from under the eyes to the lower end of the ears; lower jaw mostly white; back of the ear dark black. The color of the body and the back of the tail was similar; the tail was upturned, the upper side was black, and the black area was varying in size in different color types. All the color types exhibited a “dark top and light bottom” color transition.

The morphological characteristics of the 10 color types are described as follows:(a)Cinnamon form: Distinguished by the entire body being cinnamon (Figure 2a).(b)Brown form: Distinguished by the fur color being darker than cinnamon. The entire body is brown (Figure 2b).(c)Common form: White or yellowish-white stripes present on each of the inner corners of the eyes, and back hair turns reddish-brown. Obvious markings on the sides and back are lacking. This form corresponds to the common name and to the trade name “sesame leopard” used in early fur acquisitions (Figure 2c). This color type was the most common type in the study area, as well as in China, and it is, therefore, often referred to as the “common color type”.(d)Ocelot form: Entire body is red with red cloud-like patches darker than the body color (Figure 2d).(e)Reddish-brown long hair: The overall coat is bright red, similar to that of *Muntiacus vaginalis*, with darker extremities. No visible spots on the body. This form corresponds to the common name or trade names “red gold cat” and “red tone leopard” in early fur acquisitions. Body hair is longer, and the coat is thicker than those in other color types (Figure 2e).(f)Brown short hair: Body color is lighter orange, and body hair is shorter than that of the reddish-brown type (Figure 2f).(g)Gray form: Entire body is lead-gray, lacking markings on the whole body except the head (Figure 2g).(h)Blackening form: This color form is regarded as a gradually darkening gray form, that never reaches black. The color is intermediate between the gray and black forms, with a mosaic of black tile-like patches on the limbs around the abdomen (Figure 2h).(i)Melanistic form: The overall coat color is jet black or dark gray-black, without obvious markings. This type corresponds to the common name or the trade names “black leopard” and “clouded leopard” in early fur acquisitions (Figure 2i).(j)Pure black form: The entire body is jet black and has none of the above-mentioned common characteristics (Figure 2j).

In the surveyed population, the common form was the most dominant with 123 photos (35.76% of the 344 identifiable photos), followed by the cinnamon form (76 photos; 22.09%), reddish-brown long hair type (57; 16.57%), ocelot form (35; 10.17%), blackening form (22; 6.04%), melanistic form (8; 2.33%), gray form (7; 2.03%), brown form (7; 2.03%), brown short hair form (5; 1.45%), and the pure black form (4; 1.16%).

Black cat fur with eumelanin accounted for 11.92% and the red cat fur with phaeomelanin accounted for 88.08% of the Asian golden cat color types photographed. In natural environments, red cat fur was 7.3 times more frequent than black cat fur (Figure 3).

### 3.3. Broken Tail Phenomenon in the Asian Golden Cats

Broken tails were recorded among the common and brown short-haired individuals in the Bixiri Area. Based on the relationship between the deployment of continuous camera traps and the monitoring area, as well as the results of the second terrestrial wildlife resource survey in the Tibet Autonomous Region, the estimated average number of Asian golden cats in the Yarlung Zangbo Grand Canyon National Reserve was close to 0.08–0.12 cats per square kilometer (total: 733–1100). This is much lower than the expected population density. It is likely that high intraspecific competition and “fighting”, which is likely intensified during the estrus cycle, promoted the occurrence of tail docking in Asian golden cats. The short tail phenotype may also be attributed to a “broken tail gene” in the Asian golden cat population, but this hypothesis requires further verification (Figure 4).

## 4. Discussion

Studies on animal coat coloring and patterning increasingly use image processing techniques, such as Turing dynamics, Fourier analysis, wavelet analysis, edge detection, and combinations of these and other techniques, to obtain quantitative statistics of animal coat pattern features [28,29,30,31,32,33]. However, this approach relies on the acquisition of standardized and calibrated images. The models generated from these images have successfully replicated the patterns. However, the biological significance of these patterns remains unknown.

The body color of cats is controlled by numerous genes, some of which are antagonistic, while others are synergistic [16,18,19,20]. Allelic mutation or linkage is a natural variation that occurs due to natural genetic changes or the aggregated expression of recessive genes. Recessive genes appear at higher frequencies after inbreeding, backcrossing, and crossbreeding. White tigers and white lions are uncommon in the wild because they lack normal camouflage. In contrast, albinism is abnormally high in captivity [34]. Albino and melanization are the most common mutations in felines and are caused by the methylation of a single gene or from the interaction of several different alleles [15,35,36,37].

Melanocytes in feline skin mainly secrete two pigments: eumelanin and phaeomelanin. Eumelanin is responsible for the production of black and gray tones, that is, black pigmentation, whereas pheomelanin is responsible for the red, orange, and yellow hues of red pigmentation seen in cats. The coat colors in domestic cats are mainly caused by variation in the expression and deposition of these pigments and are controlled by the expression of specific genes such as the white spot gene *albino*, browning gene, and the *dense pigment* gene. We believe that this phenomenon is also common in Asiatic golden cats and that its molecular biological significance is similar to that in domestic cats.

Abnormal fur color in felines is often accompanied by abnormal body size, long hair, and a short or even a missing tail. Felines exhibiting these traits are less likely to survive in the wild [21,28]. In captivity, humans control the reproduction of distinctive traits and, thereby, dictate the color of many domestic cats. In the wild, natural selection weakens the survival chances of mutant animals, leading to the disappearance of albino and melanized individuals [37,38]. The exception in felines is the Asian golden cat. In this study, distinctly colored golden cat individuals were distributed in the same domain, suggesting that in the golden cat population, the color types may only reflect the differences of genetic diversity at the individual level, but cannot be used as the basis for the division of subspecies. Whether there are differential interactions and reproductive choices among individuals with different color types cannot be judged based on the data from this study. The variability of golden cat color patterns may be the result of the combined action of complex mountain conditions and special climatic conditions.

Variation in coat coloration has been reported in nearly half of the existing 40 feline species [5]. Except for the Asian golden cat and domestic cat, which are unusually rich in coat color patterns, black and albino polymorphisms are found in other feline species, such as white lion, white tiger, black tiger, black panther, albino jungle cat, and black-spotted fishing cat, among others [31,39,40,41]. These color type variations suggest that the genes encoding melanin and albino are present in every species, but their expression is suppressed by the expression of other genes.

Epigenetic variation can help organisms adapt to local environments by altering gene expression. DNA methylation is the most studied epigenetic mechanism [3,38]. Theoretically, the population spatial structure of epigenetic variation is influenced by a variety of ecological and evolutionary processes. Similar to genetic variation, epigenetic variation is subject to diffusion limitations, natural selection, and neutral drift. However, it differs from genetic variation by characteristics such as environmental induction and intergenerational reset [42,43,44]. Therefore, in areas with high levels of landscape heterogeneity, the spatial structure of population genetics and epigenetics is more complex and diverse, whereas the effects of ecological and population processes on epigenetic variation are greater and easier to visualize.

The Grand Canyon Reserve is in the southern foothills of the Himalayas in China. It has the most complete vertical gradient of the mountain ecosystems in China. Along the elevation gradient, the ecosystems change from the low-mountain semi-evergreen monsoon rainforest to mid-mountain evergreen/semi-permanent rainforest, green broad-leaved forest, subalpine evergreen coniferous forest, alpine shrub-meadow, and alpine glacial vegetation. The area is divided into seven climatic zones: the humid climate zone of the northern edge of the low mountain tropics (500–1000 m), the mountain subtropical semi-humid climate zone (1000–2400 m), the subalpine temperate semi-humid climate zone (2400–3200 m), the subalpine cold temperate climate zone (3200–4000 m), the alpine cold climate zone (4000–4300 m), and the alpine frigid climate zone (4800–7782 m) [11]. The unique ecological, climatic, landscape, and geographic features have created a distinctive topography and contributed to the extremely rich color pattern of the Asian golden cat in this area. The Asian golden cat extends south into Southeast Asia and Indonesia. However, the observed polymorphism of fur color in the Asian golden cat is unique to the Grand Canyon reserve [11,13]. Wang et al. (2021) suggested that the nature reserves, forest parks, wetland parks, and other protected areas in this region should be integrated to establish the Eastern Himalayas National Park which will provide a large area of continuous habitat. Because the proposed national park is in the border area between different countries, its establishment will also help to promote international unity, increase national pride, and maintain national security [45].

## 5. Conclusions

In this study, we used long-term camera trap technology and data visualization techniques to monitor the Asian golden cat in the Grand Canyon Reserve. We identified 10 color coat types and recorded the phenomenon of broken tails within the Asian golden cat population. Our research reveals the varying color patterns and broken tails of Asian golden cats, which lays a foundation for understanding the species. We have analyzed and discussed the formation mechanism of color type and laid the basis to extend the scope for future research on the generating mechanisms at the molecular level. The Asian golden cat is a very shy animal, difficult to observe in the wild, and its hunting is prohibited in China. As a result, molecular sampling from this species is challenging, and consequently, genetic research on this species has been sporadic. The coloration pattern of Asian golden cats in the study area varies more than those in other areas of its distribution. Whether the mechanisms of color variation are regulated by genes, environmental pressure selection, or the combination of both warrants further study.

## Figures and Tables

**Figure 1 animals-12-01420-f001:**
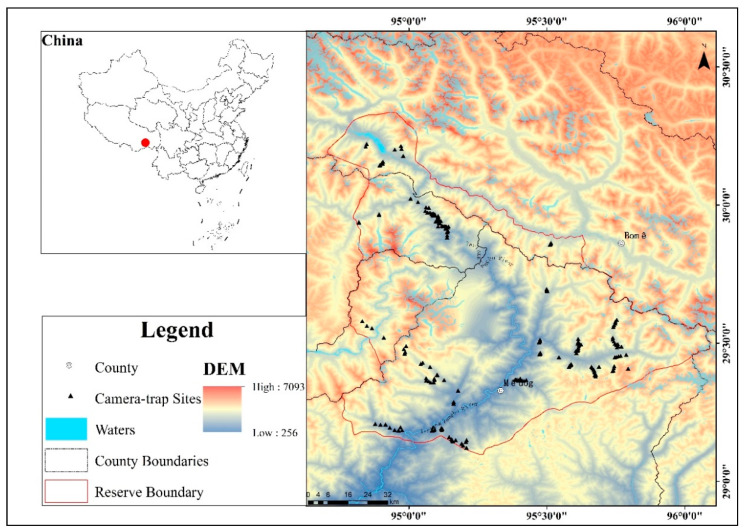
Study area and camera-trap sites.

**Figure 2 animals-12-01420-f002:**
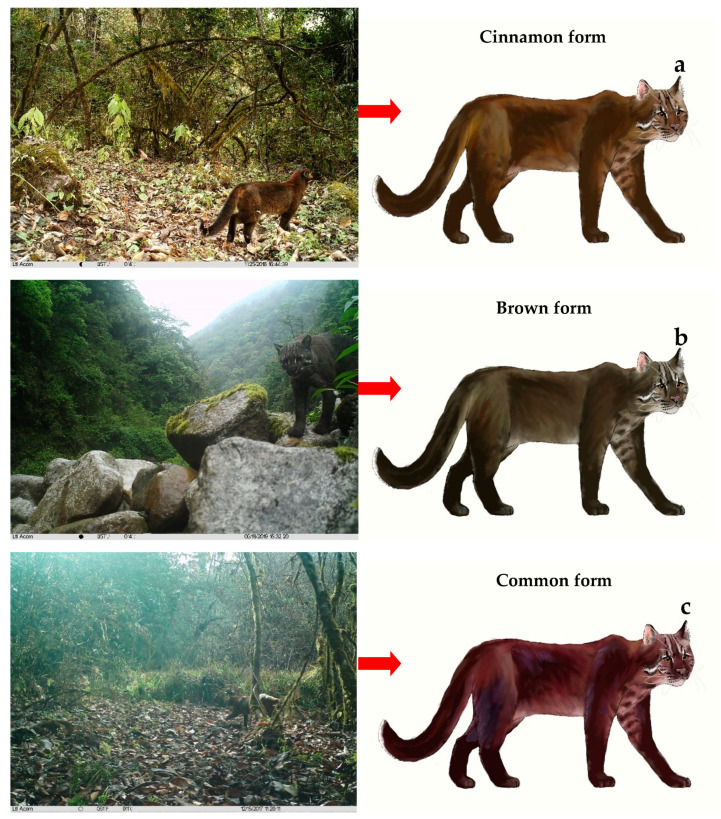
Vector images of ten coat morphs of the Asian golden cat in Yarlung Zangbo Grand Canyon National Nature Reserve, Tibet.

**Figure 3 animals-12-01420-f003:**
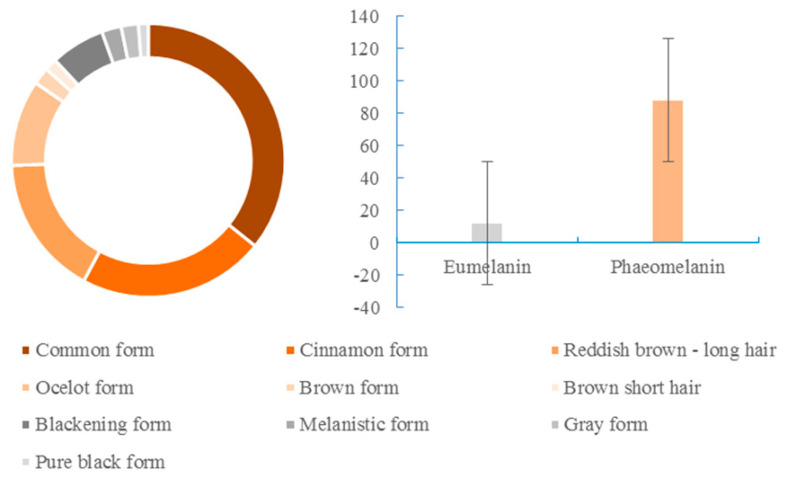
Proportion of eumelanin and phaeomelanin in natural populations of the Asian golden cat.

**Figure 4 animals-12-01420-f004:**
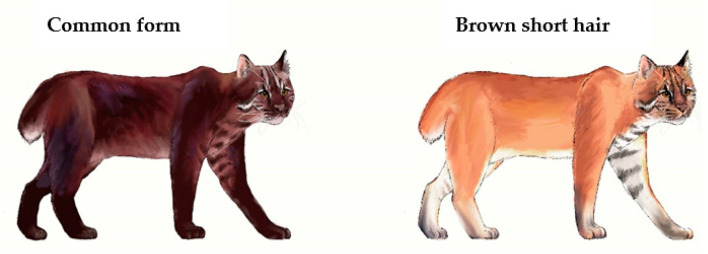
Broken tail phenomenon in common and brown short-haired forms.

**Table 1 animals-12-01420-t001:** Camera-trapping efforts and the number of independent photographs of the Asiatic golden cat in 14 survey areas for 2013–2021.

No.	Survey Areas	Number of Camera Stations	Elevation Range	Trap Nights	Number of Photographs	Number of Independent Captures of Asian Golden Cat	RAI *
1	Bixiri	11	2235–3479 m	2794	12,294	34	1.22
2	South bank of the Yarlung Zangbo River	10	582–668 m	1880	16,751	0	0.00
3	Uma Mountain	6	1751–3145 m	1374	6942	0	0.00
4	Raj Mountain	8	1631–2086 m	1968	8467	12	0.61
5	DanGeZhuo	3	954–1434 m	630	684	2	0.32
6	GeDang Ditch	27	2160–2470 m	3081	19,876	105	3.41
7	MeiYuLunBa	2	1751–2315 m	294	5172	1	0.34
8	XiGong River	6	1124–1590 m	1080	6532	3	0.28
9	GeYang Ditch	8	815–1294 m	1360	6359	3	0.22
10	DaMu	15	2001–3160 m	5850	8369	53	0.91
11	SaSong River	5	2023–2523 m	1950	8264	16	0.82
12	North of the Grand Canyon	40	1880–2980 m	14,600	20,456	55	0.38
13	DeErGong	120	1750–2890 m	45,100	38,001	201	0.45
14	Gongdui Mountain	22	2105–2780 m	8030	9080	135	1.68
	Total	283	582–3479 m	89,991	167,247	620	

* RAI is the number of independent valid photos of a species at all camera sites obtained per 100-unit camera days in a certain survey area. RAI = (number of unique valid photos/total valid camera working days) × 100.

## Data Availability

The data presented in this study are available on request from the corresponding author.

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
