# Peer review of "Recognition of Coat Pattern Variation and Broken Tail Phenomenon in the Asiatic Golden Cat (Catopuma temminckii)"

_animals, 2022, doi:10.3390/ani12111420_

Round 1
Reviewer 1 Report
The study "Recognition of coat pattern variation and broken tail phenomenon in the Asiatic golden cat (Catopuma temminckii)" is interested and contributes to a better understanding of color variability of Asiatic golden cat in natural conditions. In my opinion, the work is suitable for publication, but some elements should be improved , e.g. the description of the methods, the presentation of the results, as well as the structure of the work. Detailed comments below
18- the common form of what?
19-20 - this sentence should be moved after the sentence in line 17-18
107-112 - this part should be moved to methods section (or at least to discussion), because it describes methodological limitations.
122 - only in protected areas? why not the conservation in general?
2.1 -Study area - would be nice to see some habitat, altitudes and climate description
139- which habitats?
144- why these altitudes?
- What was the mean and minimal distance between cameras,
Fig. 1 - the figure seem to be of poor quality.
Table 1 -what is in fact survey area? I can now see, that in such area more than one trap was located. In the methods (Line 139-140) you state, that there were 11 survey areas, later (143-144) that there were 283 survey sites. This is not clear what is site, and what is area?
Besides, in my opinion the areas with Asiatic golden cats should be shown on the map
228 - remove dot before the parenthesis
253 - again dot
263-263 - not in natural environments but only in studied sites. If you disagree with me, prove that the camera traps were placed so that they represent the entire natural environment of this species' range
Fig. 3 This drawing is completely illegible due to the color scheme and the lack of reference to the charts.
278-288 - this part should be moved to discussion, results should describe your findings
299 support with citation
305 support with citation
309 support with citation
228-247 - OK, here is what I wanted to see, please move this part to the methods section
357 - national security? I agree that transboundary protection areas is a great idea, but please focus on the conservation effects on the studied species.
363: understanding the species?
Acknowledgements: I extend thanks..., I am especially... I would like... - who? Only one author, what about others?
Reviewer 2 Report
Well written and interesting paper
Intro
Line 93
However, sampling biological material from the Asian golden cat has been challenging.
Why has it been challenging, please explain.
Delete this sentence
Line 122
We also pose additional scientific questions that should be explored in future research.
methods
Change this paragraph
- When a species was photographed by an infrared camera at a single site, it was recorded as an effective detection of the species (number of independent effective photos). (2) From the first photo of this species, the photos of the same species (whether it was the same individual or not) continuously taken at this site within 30 min were counted as the same detection, that is, the number of independent effective photos. (3) The number of independent effective photos is unrelated to the number of individuals of the same kind of animals taken in a single photo or during a single detection. [11].
To this
- (1) When a Asian golden cat was photographed by an infrared camera at a single site, it was recorded as an effective detection of the cat (number of independent effective photos). (2) From the first photo of the golden cat, the photos of the same species (whether it was the same individual or not) continuously taken at this site within 30 min were counted as the same detection, that is, the number of independent effective photos. (3) The number of independent effective photos is unrelated to the number of individuals of the same kind of animals taken in a single photo or during a single detection. [11].
What other software? If you have this is methods you need to explain fully.
Line 179 were used among other software
Results
In line 186 was should be were
Adjust you table so elevation is a single line
Line 205 add were – see below
The external morphological features shared by individuals of all color types were: facial markings (add were) relatively consistent, with a wide white stripe
The back of the tail similar to body color;
Change to
The color of the body and back of the tail is similar
Line 256 delete - Only one individual was recorded each time a photo was taken.
Line 284 delete - for some areas
Reviewer 3 Report
The manuscript is interesting, and the sampling effort is huge, which is a great merit. However, some important information is missing in the methodology, which make difficult the interpretation of the results. The authors did not explain how they classified the animals into the 10 categories, if they don't explain it properly this seems arbitrary.
Consider my comments made on the manuscript (see the attached PDF), and after the correctos I can consider the work for publication, but unfortunately I cannot recommend it for publication in the current form.

Round 2
Reviewer 3 Report
The authors have improved the manuscript, I suggest to accept it.